# Reevaluating the Language of Learning Advantage in Bilingual Arithmetic: An ERP Study on Spoken Multiplication Verification

**DOI:** 10.3390/brainsci12050532

**Published:** 2022-04-21

**Authors:** Vanessa R. Cerda, Paola Montufar Soria, Nicole Y. Wicha

**Affiliations:** Department of Neuroscience, Developmental and Regenerative Biology, University of Texas at San Antonio, 1 UTSA Circle, San Antonio, TX 78249, USA; pmontufarsoria@gmail.com (P.M.S.); nicole.wicha@utsa.edu (N.Y.W.)

**Keywords:** bilingualism, arithmetic, ERPs, spoken number words, P300, N400, multiplication

## Abstract

Many studies of bilingual arithmetic report better performance when verifying arithmetic facts in the language of learning (LA+) over the other language (LA−). This could be due to language-specific memory representations, processes established during learning, or to language and task factors not related to math. The current study builds on a small number of event-related potential (ERP) studies to test this question while controlling language proficiency and eliminating potential task confounds. Adults proficient in two languages verified single-digit multiplications presented as spoken number words in LA+ and LA−, separately. ERPs and correctness judgments were measured from solution onset. Equivalent P300 effects, with larger positive amplitude for correct than incorrect solutions, were observed in both languages (Experiment 1A), even when stimuli presentation rate was shortened to increase difficulty (Experiment 1B). This effect paralleled the arithmetic correctness effect for trials presented as all digits (e.g., 2 4 8 versus 2 4 10), reflecting efficient categorization of the solutions, and was distinct from an N400 generated in a word–picture matching task, reflecting meaning processing (Experiment 2). The findings reveal that the language effects on arithmetic are likely driven by language and task factors rather than differences in memory representation in each language.

## 1. Introduction

Bilinguals are capable of learning and using math concepts in two languages. Though performing calculations or estimations are generally thought to be language-independent, retrieving memorized math facts are thought to be encoded and accessible from memory in one language, either preferentially or exclusively [1,2]. Multiplication tables, which are typically memorized as arithmetic facts in childhood, are an example of the latter. Often, the language bilinguals used when learning math, herein LA+, is preferred over their other language, or LA−, and this preference can persist into adulthood [3,4,5]. In line with this self-reported preference, multiple studies have reported that bilinguals perform or verify arithmetic faster and/or more accurately in LA+ than LA− [1,2,4,6,7,8,9,10,11,12]. This suggests that the memory for arithmetic facts might be accessible more efficiently (or directly only) in LA+.

However, some studies suggest that differences in arithmetic performance across languages may be due more to individual differences in language proficiency or language use than memory encoding differences specific to arithmetic [13,14]. The format in which math facts are presented—digits, written, or spoken number words—and the task demands may also contribute to observing these differences in processing [6,15]. The current study builds on a small number of studies investigating the neural time course for multiplication verification in the bilingual brain [4,6,14]. The primary goal is to understand whether arithmetic representations are accessed preferentially or exclusively in one language. We do this by resolving discrepancies in methodology across previous studies and controlling for linguistic and methodological factors. We briefly introduce the relevant theoretical models and prior research to place our predictions in context.

The two leading models of cognitive arithmetic are the Encoding-Complex Model (ECM) [16] and the Triple Code Model (TCM) [17]. ECM accounts for performance differences in a spreading activation network with memory representations, or lexical stores, for arithmetic facts and numbers in each language [16]. These representations are separate from each other (and from the representation for digits), and from the magnitude representation to which all of these stores are connected. ECM is the only model that explicitly considers a bilingual’s experience. It attributes differences in performance with memorized arithmetic facts across languages to differences in strength in access to the facts (e.g., weighted connections in the memory network) rather than different processing strategies altogether [16]. The strength of the connections between facts, both within a language and between each lexical store and other parts of the network, can be modulated by the relative experience of using arithmetic in that language. In turn, retrieval of arithmetic facts from memory are most efficient when problems are presented in the most commonly used visual digit format (e.g., Arabic numerals), followed by LA+ and then LA−.

The second model, TCM, does not explicitly address bilingual arithmetic, but it has been used to support language specific encoding of arithmetic facts [2,17]. TCM proposes that multiplication facts are represented as a learned lexicon of auditory verbal associations and suggests that this verbal code exists only in the language of learning [1,2]. Similar to ECM, the verbal code is connected to a separate magnitude code representing quantity or numerical value. Neuroimaging studies have provided support for the existence of separate brain areas engaged in the proposed verbal and magnitude codes [1,18,19]. Given that arithmetic facts are represented solely in LA+, bilinguals must engage different cognitive strategies in each language. Retrieval of arithmetic facts from memory would only be possible in LA+, the language in which arithmetic facts were originally encoded. Any other language or format, including the written digit format, would require translation into LA+.

Event-related potentials (ERPs) have provided a window into the cognitive processes underlying multiplication verification in both monolinguals and bilinguals [4,6,14,15,20,21,22,23,24,25]. ERPs are a measurement of the synchronized firing of cortical neurons with millisecond precision. This electrophysiological measurement at the scalp can detect the unfolding of different cognitive events between experiencing a stimulus and making a behavioral response. Previous studies have used ERPs to measure the neural time course of bilingual arithmetic to shed light on whether bilinguals retrieve memorized math facts from memory either preferentially or exclusively in one language. These studies have not supported the hypothesis that bilinguals engage different cognitive processes in each language, as suggested by TCM. However, they have supported ECM’s predictions of less efficient processing in LA−.

Salillas and Wicha [4] were the first to use ERPs to examine how bilingual adults verify multiplication facts in each of their languages. Multiplication problems were presented as three sequential written number words in English or Spanish (e.g., two five ten; dos cinco diez), and participants were asked to verify by button-press whether or not the third number was the correct product for the first two operands. The participants were considered early bilinguals (learned both languages before 6 years of age) and balanced (equivalent proficiency in both languages) but learned and preferred to use arithmetic in their LA+ [see 4]. The results showed similar timing and polarity of the ERP effect in both LA+ and LA−, suggesting the engagement of similar cognitive processes in both languages. However, LA+ elicited a larger ERP effect (difference in amplitude between correct and incorrect solutions) than LA-. The authors concluded that even with equivalent proficiency in their languages, bilinguals have more efficient or more refined access to the memorized multiplication facts when using LA+ than when using LA− (The authors also manipulated whether incorrect solutions were related or unrelated to the preceding operands and reported that a difference in relatedness was observed only in LA+. However, this effect is elusive and has not been replicated in our lab). This suggested the presence of stronger memory networks for arithmetic in LA+.

Martinez-Lincoln, Cortinas, and Wicha [14] later used the Salillas and Wicha [4] paradigm with bilingual primary school educators and observed that the difference in the correctness effect between languages could be mitigated by using LA−. Educators who taught in their LA+ showed a larger correctness effect in LA+ than LA−, as in the original Salillas and Wicha [4] study. However, educators who taught in their LA- showed equivalent ERP correctness effects in both languages. In other words, language use modulated the efficiency of accessing math facts in the LA−, which suggests that the memory networks for arithmetic facts are flexible and sensitive to experience. This is in line with the proposed associative networks of ECM, where the strength of connections between arithmetic facts and between the language stores can be modulated by use [16].

More recently, Cerda et al. [6] showed that early bilingual children also exhibit similar ERP correctness effects in both of their languages. In this study, Spanish–English bilingual elementary school children (3rd to 5th grade) performed a similar multiplication verification task as in Salillas and Wicha [4], but with some methodological differences discussed below. Like the adults in the above-mentioned studies, the children learned both languages early in life and had equivalent proficiency in both languages but learned multiplication in one language. Surprisingly, in contrast to the findings in adults, the ERP correctness effect was not different between languages, even though the children recently learned and performed multiplication in only one of their languages. Based on this finding, the authors hypothesized that the language effect observed in bilingual adults might be attributed to the continued use of LA+ over a lifetime, rather than a difference that is established during early learning, as suggested by Salillas and Wicha [4].

However, there were some methodological differences that might explain why adults, but not children, showed a larger effect of correctness in LA+ than LA−. First, to avoid a confound of reading ability with children, Cerda et al. [6] used a novel paradigm where the first two operands were presented as spoken number words in English or Spanish on separate trials. The solution was always presented as a digit (e.g., “two” “three” 6). In this way, effects at the solution would reflect access to the multiplication facts based on the preceding spoken operands, and not lexical differences at the solution itself. Moreover, attenuated ERP effects have been observed for monolingual children who are low-ability readers [26] or have developmental dyslexia [27] compared to high-ability readers. Similarly, reading in a language that is not typically used for reading (e.g., heritage speakers) could also influence multiplication verification when the problems are presented as written number words in that weaker language. Both written and spoken number words engage one language explicitly while accessing the multiplication facts from memory, but spoken words are a more natural way of performing single-digit multiplication problems, see [28].

A second difference between the studies was that the languages were blocked in the child study, such that they performed the task in one language at a time. In contrast, the stimuli in the adult study were mixed with consecutive trials appearing randomly in either English or Spanish. Bilingualism research has shown that switching between languages causes asymmetric interference effects in a dominant versus non-dominant language [29,30,31,32]. In turn, the adult bilingual task may have elicited differences across languages, not because of differences in access to the multiplication facts, but because of asymmetric language switching effects. In fact, one study investigating language preference in simple arithmetic showed that a preference for the LA+ only occurred when the trials switched between languages, but not when the trials were blocked by language [33]. An asymmetric switching effect has also been observed for number word reading in bilingual adults [34].

So far, we have not revealed the specific ERP component modulated by correctness in the above-mentioned studies to emphasize the point that the same brain response was always observed in both languages. This was true for both children and adults. However, the type of ERP response observed is critical for elucidating which cognitive processes support arithmetic in the bilingual brain. All three of the studies discussed above revealed an effect of correctness where solutions elicited a negative-going deflection, called the N400, with less negative amplitude for correct than incorrect solutions. This shift in negative amplitude for the N400 typically begins around 250 ms from the onset of any meaningful or potentially meaningful stimulus, and peaks around 400 ms [35,36]. The N400 can be thought of as an index of the current state of activation in semantic memory. When reading a sentence, for example, words that are expected based on preceding context will elicit greater activation in semantic memory, and in turn, smaller N400 amplitude than words that are not supported by context [35,37,38]. Under an N400 interpretation of the arithmetic effect, correct solutions are more facilitated in memory after reading the first two operands, and therefore elicit less N400 amplitude compared to incorrect solutions [23,24,25].

Salillas and Wicha [4] and Martinez-Lincoln et al. [14] both showed a correctness modulation of the N400 for the multiplication problems that were presented as written words in LA+ and LA− (Salillas and Wicha [4] also reported an N400 to problems presented in the digit format, but this has been reinterpreted as a P300, as discussed herein). The larger N400 effect in LA+ was interpreted as indicating that, although semantic memory was accessed to judge the correctness of the problems in both languages, there was a difference across languages in the efficiency of access (and in the spread of activation). In other words, the memory network for LA+ was stronger than LA−, especially when LA+ continued to be the preferred language of use into adulthood [14]. For adults who practiced accessing these facts from memory in LA−, like the bilingual educators in the Martinez-Lincoln et al. study, the memory networks for both languages became equally efficient. Children also exhibited a robust N400 effect with smaller amplitude for correct solutions, comparable to an N400 elicited in a language task (i.e., word–picture verification) in the same children [6]. This N400 correctness effect was not different in timing, amplitude, or distribution across the children’s two languages.

These studies with bilinguals address a broader body of literature that investigates the cognitive processes underlying multiplication verification in the adult brain. Niedeggen and colleagues were the first to report ERP measures from adults verifying simple multiplication problems [23,24]. In these studies, participants saw three consecutive Arabic numbers (digits) and judged if the third number was the correct product of the first two (e.g., 2 4 9). The findings revealed a robust effect of arithmetic correctness, about 250 ms after the third number appeared, with incorrect solutions eliciting more negative amplitude compared to correct solutions. Based on the hypothesis that math facts should be stored in verbal memory, this arithmetic correctness effect was interpreted as a modulation of the N400 [23,24].

In addition to the trials presented as written numbers, Salillas and Wicha [4] also included a condition with all digits and observed an effect similar to Niedeggen and colleagues [23,24]. Given the precedent for interpreting the effect as an N400, Salillas and Wicha adopted the N400 interpretation for their findings in the digit trials. This was a parsimonious interpretation of the findings for both digit and number word trials. However, several researchers have reevaluated the underlying ERP componentry, and in turn, the cognitive processes, driving this arithmetic correctness effect. Jasinski and Coch [22] were the first to point out that the N400 interpretation of the arithmetic correctness effect was an error. Because ERP effects are measured as a relative difference in mean amplitude between two conditions, and not as absolute amplitude values, the correctness effect could either be interpreted as an increase in negative amplitude for incorrect solutions, as per Niedeggen and colleagues, or an increase in positive amplitude for correct solutions. Using an arithmetic verification paradigm similar to Niedeggen and Rosler [23], Jasinski and Coch [22] observed the same pattern where incorrect solutions elicited more negative amplitude compared to correct solutions. However, they observed that the distribution and timing of the effect was not typical of an N400 and instead likened the positive deflection for correct solutions to a target P300 (or P3b), reflecting retrieval of correct solutions from working memory.

The P300 is a positive-going deflection that is elicited when a subject makes a categorical decision about a given stimulus [39,40,41,42]. The P300 is larger in amplitude for anticipated, task-relevant targets than for distractors [43], and for items that are more easily classified relative to items that are harder to classify, e.g., [40]. Under this explanation, the arithmetic correctness effect is driven by a larger P300 to the correct solutions and reflects easier categorization of correct (target) solutions for highly rehearsed arithmetic facts, rather than differential access to semantic memory as would be implied by an N400. This P300 interpretation of the ERP correctness effect in adults has been supported in multiple subsequent studies [15,20,21,44].

Indeed, when comparing the ERP waveforms from these more recent studies [15,20,21,22,44] to the original reported “N400” correctness effect in the digit format [4,23,24], it is clear that the morphology of the waveforms is quite similar to P300. When Salillas and Wicha [4] used the digit format, the correctness effect in bilingual adults, which they called an N400, looks nearly identical to other P300 effects [15,20,21,22,23,24], with a positive going amplitude modulation for correct solutions.

This does not mean that an arithmetic correctness effect is never observed on the N400. In fact, multiple studies have shown that children exhibit an N400 correctness effect both for digit and number word stimuli [6,44,45]. In a direct comparison between children and adults, Grenier et al. [44] showed that the adult arithmetic correctness effect in response to trials presented as digits was distinct from the N400 effect in children when performing the same task. The adult response was also distinct in distribution and morphology from an N400 effect that was generated in a word–picture verification task in the same adults [44]. Similarly, the effect in Cerda et al. [6] for bilingual children was also an N400 comparable to an N400 elicited to a language task.

As for adults, specific experimental manipulations seem to promote the presence of an N400 correctness effect for multiplication verification. In Dickson et al. [15], adult participants were asked to withhold their correctness judgment until a response prompt appeared one second after the presentation of the solution. The motivation for this was to avoid motor artifact in the ERPs to the solution, but this delay seems to encourage the engagement of semantic level processing. They found that using spoken operands in combination with a delayed response to a digit solution elicited an N400 effect. Similarly, Salillas and Wicha [4] and Martinez-Lincoln et al. [14] required a delayed response for trials presented as written number words. These trials clearly elicited a negative-going deflection with increased negative amplitude for incorrect solutions, or an N400 correctness effect. However, the same task performed with digit operands elicited a P300 [4,15]. Moreover, requiring immediate responses upon the presentation of the solution leads to a P300 response even for number words [21]. Therefore the critical combination for eliciting an N400 is problems presented as words with a delayed response, as in Salillas and Wicha [4].

So, what does this reinterpretation of the arithmetic correctness effect mean for the bilingual adult studies? The current study aims to address this question and resolve the discrepancies in the ERP results across this small number of arithmetic experiments, as described below.

### The Current Study

We aimed to address the methodological differences between previous bilingual arithmetic ERP studies as a means to further examine how bilingual adults process multiplication facts in each of their languages. Using the Cerda et al. [6] paradigm with spoken multiplication problems and an immediate correctness judgment, we tested for differences in access to multiplication facts across a bilingual’s languages. Although previous adult studies used written number word stimuli, arithmetic facts are rarely encountered as written words and are processed less efficiently compared to more familiar Arabic digit stimuli [28]. By using spoken number words followed by a digit solution, we can measure differences in processing in each language while avoiding confounds of reading fluency that might be present for written words, as well as lexical differences at the solution itself. This is important for understanding whether adult bilinguals indeed access these multiplication facts from memory differentially across their languages, or if previously reported differences might be explained by the methodology used.

Moreover, by measuring simultaneous behavior and ERPs, we will determine which cognitive process might account for performance differences between LA+ and LA−. We will determine if the ERP correctness effect is a modulation of the N400, as observed by Salillas and Wicha [4] using written number words in a mixed language context and a delayed response, or a P300 response like that observed in adults on a similar spoken number word task [21,44].

Our findings can speak to the leading models of arithmetic. Based on TCM, bilingual adults should engage different cognitive strategies in each language when processing multiplication facts, namely, memory retrieval when using LA+ versus translation or computation when using LA− [1,2,17]. By inference, TCM would predict that bilingual adults should show a brain response similar to monolingual adults only in the LA+, with a different brain response altogether in LA−. For example, we might expect modulations of a P300 in LA+ and a childlike N400 in LA− [44]. Although Salillas and Wicha [4] concluded that adults were employing the same cognitive strategies across languages for arithmetic, the use of written number words with delayed responses may have promoted an N400 correctness effect in both languages, as observed with monolingual adults [15]. 

In contrast, ECM suggests that both LA+ and LA− should have similar but independent representations of the multiplication facts in each language. The model proposes that bilinguals should be more efficient at accessing multiplication facts when using LA+, but also has an allowance for frequency of use to strengthen connections in each language. We therefore infer that LA+ and LA− should show similar brain responses, whether it be a P300 or an N400. If the original language of learning elicits more efficient access to the math facts, this would result in a faster or larger ERP effect in LA+ than LA− (e.g., larger P300 to correct trials or larger N400 to incorrect trials in LA+ than LA−, depending on the ERP component observed).

One challenge with both of these models, and the bilingual arithmetic literature more broadly, is that little is said about how the lexicon (or lexica) of memorized arithmetic facts relates to a bilingual’s languages, or language experience. Factors associated with language experience, like language fluency, frequency of use, and age of acquisition can influence how verbal information is processed [46,47,48]. For example, age of acquisition and language proficiency can modulate the degree of overlap in the brain areas activated for each of a bilingual’s languages [31,49]. Age of acquisition and vocabulary proficiency affect the speed at which words are integrated into a semantic context when bilinguals read sentences across their languages [50].

If arithmetic facts are indeed stored in a verbal code, it is possible that these language factors could impact the efficiency of access to arithmetic facts, and potentially explain previous reports of the LA+ advantage for arithmetic. Tamamaki [10] proposed decades ago that the language bias for arithmetic was dependent on a bilingual’s L2 proficiency. Bilinguals with low proficiency in their L2 were overall faster at producing solutions to arithmetic problems presented in their first language, supporting a language bias for arithmetic. However, bilinguals with high proficiency in L2 showed reaction times in L1 and L2 that were virtually identical. Moreover, as bilinguals increased their use of L2, ease of arithmetic processing in L2 increased as well. Similarly, the teacher study discussed above revealed that language use can override the LA+ advantage [14], and a recent meta-analysis investigation strongly suggests that language differences in processing arithmetic can be explained by these language use factors [13]. Still, these critical factors are not always considered in studies of bilingual arithmetic [1,2,7,8,9,11,19,51].

The current study carefully controlled for these language factors. We used a set of objective and self-report language measures to reduce the heterogeneity of bilingual factors that could explain differences across languages in the arithmetic correctness effect. This is critical to understanding if any differences observed are due to memory encoding, retrieval, or other processes engaged during arithmetic verification without the potential confounds of linguistic factors. Our sample includes early Spanish–English bilinguals who learned both languages before learning early arithmetic facts, similar to the children included in Cerda et al. [6]. In this way, we could dissociate the role of the language of learning from the age of acquisition of that language, which other studies did not control [1,2,7,8,11,51,52,53]. In addition, all participants had equivalent fluency in both languages and frequently used both languages in daily life, yet learned multiplication primarily or solely in one language.

Given the recent reevaluation of the arithmetic correctness effect in adults and the methodological differences across previous bilingual arithmetic studies, the current study aimed to address three questions. First, we aimed to determine if the ERP correctness effect in bilingual adults is a modulation of the N400 or the P300. In Experiment 1A, multiplication trials were presented as three consecutive Arabic digit numbers, similar to Dickson et al. [15,21] and Salillas and Wicha [4]. ERPs and behavior (response time and accuracy) were measured from solution onset (not delayed). Following previous studies, we expected that these experimental conditions would promote modulations of the P300.

Second, we aimed to determine whether the correctness effect differs between spoken number word and digit trials. Experiment 1A included trials in which adult bilinguals heard two number words in either Spanish or English in separate alternate blocks, then saw an Arabic numeral solution. This paradigm was adapted from Cerda et al. [6], and allowed for direct comparison across formats to determine if the digits and spoken word trials elicit different ERP components in bilinguals, as in Salillas and Wicha [4]. In Experiment 2, we used a language task known to generate a robust N400 [44]. In this task, the same bilingual participants judged the (mis)match of a spoken word and picture in each language, separately. This task served as a point of comparison across languages not specific to multiplication processing and helped to confirm whether the arithmetic correctness effect for spoken word trials elicited modulations of an N400 or a P300.

Third, we aimed to determine if there is a difference in the correctness effect between LA+ and LA− (i.e., evidence of a language bias) in terms of amplitude, polarity, or distribution. Experiment 1A used the paradigm from Cerda, et al. [6]. This is the first ERP study to measure bilingual arithmetic processing in each of a bilingual’s languages using spoken number word trials in adults. It was important to use the original paradigm so that we could directly compare the adult response to previous work in children. Given that we used the paradigm that was designed for children, to test the robustness of the results from Experiment 1A, Experiment 1B increased the task demands by speeding up the presentation of the stimuli. 

Together, these findings will help determine the cognitive processes underlying arithmetic verification in the bilingual adult brain.

## 2. Materials and Methods

### 2.1. Experiment 1: Multiplication Verification Task

#### 2.1.1. Participants

Experiment 1A included 29 balanced Spanish–English bilingual adults (19 female; mean age 20.7 years; range 18–26 years). Experiment 1B included 28 participants (15 female; mean age 21.25 years; range 18–26 years). Twelve participants completed both experiments (average time between experiments: 16 months; range: 12–20 months). Additional data from 6 participants from Experiment 1A and 2 participants from Experiment 1B were excluded for either having too few trials in the conditions of interest or not being equally proficient across languages at the time of testing. Participants were all right-handed (as assessed by the Edinburgh Inventory: Experiment 1A average: 0.86; range: 0.5–1.0; Experiment 1B average: 0.83; range: 0.5–1.0) [54] undergraduate students from the University of Texas San Antonio.

All participants were considered balanced Spanish–English bilinguals, where language proficiency was based on comparisons of standardized test measures in each language, as described below. All subjects reported Spanish as their native language (L1), with the exception of 1 participant in Experiment 1B who reported learning both languages in parallel since birth. Notably, L1 was not always the language of learning for multiplication: 18 of 29 participants in Experiment 1A and 14 of 28 participants in Experiment 1B reported learning multiplication in English (L2). For Experiment 1A, 17 participants reported preferring to use their LA+ for multiplication at the time of testing, 4 reported having no language preference for multiplication, and 4 participants reported preferring their LA−. For Experiment 1B, 12 participants reported preferring to use their LA+, 13 reported having no language preference, and 3 participants reported preferring their LA−. Note that all participants who preferred using their LA− for multiplication reported that their LA− was English.

Participants had no history of cognitive or perceptual (visual or auditory) deficits, had normal or corrected-to-normal vision, and were not on any medication routines that affected their cognition at the time of testing. No participants had been diagnosed with language delays or learning disabilities. Informed consent was received in accordance with the Institutional Review Board (IRB) from the University of Texas at San Antonio.

#### 2.1.2. Offline Behavioral Assessments

To determine proficiency across both languages, two subtests (Test 14: Picture Vocabulary and Test 15: Oral Comprehension) of The Woodcock–Johnson Tests of Achievement [55] and one subtest (Test 8: Incomplete Words) of the Woodcock–Johnson Tests of Cognitive Abilities [56] were used. Equivalent versions of these tests in Spanish from the Batería III Woodcock–Muñoz (Batería III) [57] were also administered. These tests required subjects to name pictures (Test 14), complete oral sentences (Test 15), and demonstrate phonological awareness (Test 8) in each language. Age-normalized classifications were obtained from the Woodcock–Johnson Proficiency Battery (WJPB-R) based on a subject’s raw assessment scores in each language. These classifications were then compared across each of the parallel subtests from the WJPB-R and the Batería III. For inclusion in the study, participants were considered balanced if their proficiency scores for each subtest were within 1 +/− classification across languages for at least 2 out of the 3 tests. Table 1 shows the mean (SD) for these tests. Language and math history was also collected using an adapted version of the Language Experience and Proficiency Questionnaire (LEAP-Q) [58]. Subjects’ working memory and math fluency were assessed using the Numbers Reversed subtest of the WJ III COG and the Wechsler Individual Achievement Test (math fluency-multiplication, WIAT III) [59], respectively.

#### 2.1.3. Stimuli

Stimulus materials were identical to Cerda et al. [6], consisting of all single-digit multiplication problems with operands ranging from 2 to 9. Operands were presented as Arabic digits, spoken number words in English, or spoken number words in Spanish in separate blocks, always followed by a visual Arabic numeral solution. Using digit solutions allows for direct comparison across languages, see [15]. After the elimination of tie problems (e.g., 2 × 2, 3 × 3, etc.) and problems containing 0 and 1, see [60,61], a set of 28 core problems remained. Each problem was repeated four times, twice with a correct solution (e.g., 2 × 4 = 8 and 4 × 2 = 8) and twice with an incorrect solution (e.g., 2 × 4 = 12 and 4 × 2 = 10), creating a total of 112 problems presented in each language and the digit format. All incorrect solutions were a multiple of either the first or second operand (never more than 3 multiples larger or smaller), making them all table related. We did not include table unrelated solutions because the reproducibility of the relatedness effect even in monolingual adults remains controversial, with nuances of stimulus design as well as important differences in component identification across studies [20,23]. Trials were pseudorandomized separately for each format, so that no three incorrect or correct problems were presented in a row.

Spoken operands were recorded by a female, fluent Spanish–English bilingual using a natural rate of speech. Sound files were normalized for volume and cropped to the sound onset/offset, avoiding distortions (e.g., clipping). The spoken number word sound files were then normalized to 450 ms duration chosen specifically to reduce distortion in both languages, see [6]. Spanish tends to have longer words than English, and this is true for number words as well (operand word length in letters, English range: 3–5; median: 4; Spanish range: 3–6; median: 4.5). Spanish also had more 2 syllable words than English, but critically the maximum number of syllables was 2 for both languages. This difference in word length could theoretically make processing multiplication problems slower in Spanish than English. Despite these differences, we are confident that this is not a confound in our results (see Section 3.4.1).

#### 2.1.4. Experiment 1A Procedure

In a single session, all participants performed the multiplication tasks in separate blocks of Digit, English, and Spanish trials, with alternating language order and format order (digit operands vs spoken operands) across participants. During the digit operand task, instructions and feedback were given in English. Within each language block, subjects were given instructions and feedback in the same language as the corresponding task, see [30]. Each task consisted of eight blocks of fourteen trials each. Participants were encouraged to respond as quickly and accurately as possible, and to avoid blinking or eye movements away from a center fixation. Immediately after completion of each block, they were given feedback about how many total trials they responded to correctly and were permitted to take a self-timed break to avoid fatigue.

Multiplication problems were presented as three consecutive numbers, one number at a time with no symbols between operands (e.g., 2 4 8). The first two numbers were presented either as Arabic Digits or spoken number words in the single language (450 ms duration; 250 ms interstimulus interval—ISI). Sounds were played through EEG-compatible insert headphones (ER1; Etymotic Research Inc.; Elk Grove Village, IL, USA). As the spoken operands were presented, a gold coin with an embedded multiplication symbol remained in the center of the screen. The coin served as a fixation point to prevent eye movements during auditory stimulation (see Figure 1 for an example of presentation and timing). The second operand was always followed by a 1000 ms ISI, then an Arabic digit solution appeared on the screen, replacing the fixation “coin”. Solutions were presented in white text on a black background in the center of a 19” LCD Dell monitor.

Participants were asked to indicate solution correctness as soon as possible after viewing the solution by pressing one of two buttons on a game controller (Logitech Gamepad F310) with their index fingers. Finger-response mapping (left/right hand) was reversed for half of the participants. Subjects had a maximum time of 5000 ms to record a response.

#### 2.1.5. Experiment 1B Procedure

Only the spoken multiplication trials in English and Spanish were included in Experiment 1B. The same stimuli for these trials were used as in Experiment 1a, with two modifications to the timing of the experiment. The interstimulus interval (ISI) between the first and second operand and the ISI between the second operand and the solution were removed. This change decreased the total duration of each trial (see Figure 1 to compare timing). All participants completed the multiplication verification task in English and Spanish, in separate blocks, in a single session while EEG and performance (accuracy and RT) were measured. Language order alternated across participants to control for order effects.

### 2.2. Experiment 2: Word–Picture Verification (WPV) Task

#### 2.2.1. Participants

Immediately following the multiplication task in one of the languages in Experiment 1A, all 29 participants performed a word–picture verification task (WPV) in the same language. Similar to Cerda et al. [6], participants performed this task to generate an N400 effect, an index of semantic-level processing, thereby enabling comparison of this effect across languages. Of these 29 participants, 2 additional participants were excluded from the word–picture verification task grand average due to inadequate trial counts (following the criteria in Experiment 1A).

#### 2.2.2. Stimuli and Procedure

Participants were tasked with verifying the semantic fit of word–picture pairs. A prerecorded spoken noun in English or Spanish (word duration varied, English average word length: 645 ms; range of 410–930 ms; Spanish average word length: 643 ms; range of 390–930 ms) was presented through EEG-compatible insert headphones. As the words were presented, a fixation coin was present on the screen to prevent eye movements during auditory stimulation. This was followed by a line drawing of an object (ISI 250 ms) presented in white on the center of a black background for 600 ms (images modified from [62]). The spoken word was either the most common name for the pictured object (congruent; based on [63] for English; [64] for Spanish) or a semantic anomaly (incongruent). All words were matched for frequency (using the CELEX for English words and LEXESP for Spanish words) and number of syllables (average: 2; range 1–4) across languages. Additionally, incongruent trials were matched for animacy (i.e., animate versus inanimate objects) and were different across languages to avoid predictability.

Participants were asked to identify congruent and incongruent word–picture pairs as quickly and accurately as possible by pressing one of two buttons on a game controller (Logitech Gamepad F310) as in the multiplication task. The task was divided into four blocks with 20 trials each. Trials were pseudorandomized so that no three match or mismatch trials were presented in a row. Two lists of stimuli were created per language where match and mismatched trials were reversed. These lists were alternated across participants, such that all pictures and words appeared in the math and mismatch conditions across two participants.

### 2.3. EEG Recording

Across both experiments and all tasks, participants sat in a dimly lit, sound-attenuating chamber while wearing a custom electrode cap fitted with 26 Ag-AgCl sintered electrodes (Electro-Cap International Inc.; Eaton, OH, U.S.A; using BioSemi ActiveTwo electrodes). Electrodes were geodesically arranged on the head. Continuous EEG was recorded using BioSemi ActiveTwo bioamplifier running ActiveView software. A signal from each electrode was recorded with respect to a common mode sense/driven right leg (CMS/DRL) active/ground electrode and was referenced offline to the average of two electrodes placed over the right and left mastoid processes. Noise induced by the amplification and digitization system was reduced by buffer amplifiers within the electrodes allowing for measurements of potentials from the surface of the skin with high electrode impedances (BioSemi ActiveTwo, BioSemi B.V., Amsterdam, Netherlands). Electrode offsets were kept below 50 mV. A fixed first order analog antialiasing filter with a half-power cutoff at 3.6 kHz was applied (see https://www.biosemi.com (accessed on 27 May 2021)).

The data were sampled at 256 Hz (2048 Hz with a decimation factor of 1/8) using a 5th order sinc low-pass digital filter to remove high frequencies. ERPLab software was used to process and take measurements of the EEG data [65]. Raw EEG data was measured in 1-s epochs, −100 to 900 ms, time-locked to the onset of the solution (multiplication task), or the line drawing (WPV task). Thresholds for artifact rejection algorithms were calibrated for each participant separately through visual inspection of the data, and the algorithms were then applied to the entire data set to exclude epochs containing artifacts using built-in EEGLab artifact tests. Separate moving window peak-to-peak tests were used to identify trials with blinks (Threshold range: 100–125 μV) and excessive muscle artifacts and EEG drift (Threshold range: 100–125 μV). A step function test was applied to capture horizontal eye movement (Threshold: 100 μV). After removing trials with artifacts, EEG epochs were averaged by condition, and 2nd order Butterworth digital filters with a low cutoff at 0.1 Hz and a high cutoff at 30 Hz were applied before analysis. SPSS was used to perform all statistical analyses of the ERP data.

## 3. Results

We first present the analysis and results from the multiplication task in Experiment 1A comparing across the three operand formats (digits, LA+, and LA−). This includes analysis of performance measures on the task, i.e., accuracy and response times, and analysis of the P300 time window for ERP data analysis. For completeness, we also included separate analyses of three time-windows before and after the P300. We then presented separate analyses and results from the multiplication task in Experiment 1B for the two languages, with a direct comparison of these measures across both 1A and 1B. Finally, we presented results for the WPV task to determine if the brain response in the multiplication task is an N400 or a P300, as compared to this non-math linguistic task.

Mean amplitude was measured at the P200, P300, and post-P300 window(s). Each window was determined independently using peak latency analyses. The P200 window captured format effects (digit minus spoken word trials). The P300 window captured correctness effects and shifted in time with the manipulation of ISI. Therefore, separate latency analyses determined the peak of the P300 effect, then a 200 ms window was centered on this peak. The post-P300 analysis was conducted on two consecutive time windows in Experiment 1A. Because of the shifted P300, Experiment 1B had a single post-P300 window, which overlapped with Experiment 1A. Separate ANOVAs were performed with two levels of Correctness (correct, incorrect), 3 levels of Format (digit, LA+, LA−), and 26 levels of Electrode for each time window. For completeness, any interactions found by Electrode were followed up with a distributional analysis replacing Electrode with 2 levels of Laterality, 2 levels of Hemisphere, and 4 levels of Anteriority. This distributional analysis included a standard set of electrodes from the geodesic array that allows for equal number of electrodes for each of these combinatorial contrasts, see [6]. The specific electrodes included in the distributional analyses are highlighted in grey in the headplot of Figure 2. Partial eta squared is reported for effects that reached significance.

### 3.1. ISI Multiplication Task—Experiment 1A

#### 3.1.1. Behavior

Factorial analysis of variance was used to analyze accuracy and response times separately, measured from the onset of the solution. Each 2 × 3 ANOVA had two levels of Correctness (correct, incorrect) and three levels of Format (digit, LA+, LA−). For accuracy, there was no significant effect of Correctness (correct 91%; incorrect 90%), possibly due to a ceiling effect. There was a significant effect of Format (F(1,28) = 6.48; *p* = 0.005; η^2^ = 0.324), with more accurate responses overall when problems were presented as digits (92.11%; SE = 1.207) than words (LA + SE = 1.76, t(28) = 3.43; *p* = 0.002; LA− SE = 1.62, t(28) = 3.32; *p* = 0.003). Accuracy was not different between LA+ (89.22%) and LA− (89.75%) (t(28) = −0.77; *p* = 0.48), and there was no interaction between Correctness and Format. 

For response time, only accurate responses were included in analysis. There was a main effect of Correctness (F(1,28) = 78.18; *p* < 0.0001; η^2^ = 0.736), with faster responses for verifying correct (880 ms; SE = 59.94) than incorrect solutions (1021 ms; SE = 69.371). There were no main effects of Format or interaction between Format and Correctness on response time (see Figure 3).

#### 3.1.2. ERPs

ERPs were measured from the onset of the Arabic digit solutions in all three multiplication tasks (digit, LA+, and LA−). ERPs were averaged separately for correct and incorrect solutions; only trials that subjects judged accurately were included in analyses. Subjects were excluded from the grand average if they had less than 20 trials in any of the critical conditions to ensure an adequate signal-to-noise ratio in extracting the ERPs. The average number of trials was similar across languages and conditions (digit correct—mean: 45, range: 25–55; digit incorrect—mean: 45, range: 20–56; LA+ correct—mean: 43, range: 29–55; LA+ incorrect—mean: 43, range: 27–55; LA− correct—mean: 46, range: 24–56; LA− incorrect—mean: 45, range: 29–56).

Visual inspection of the grand average ERPs revealed typical sensory components, N1-P2, across the head in response to the Arabic digit solutions for all conditions. A large difference was apparent across formats at the P200 starting around 200 ms after solution onset. Around 250 ms after solution onset, a widespread positive-going deflection is observed with larger amplitude for correct than incorrect solutions, a P300 correctness effect (Figure 4). This effect at the solution was present regardless of the format of the preceding operands including problems in both LA+ and LA−. This effect was followed by a later effect where digit trials elicited an overall more positive response compared to the spoken word trials.

#### 3.1.3. P200 Time Window (200–270 ms)

Between 200–270 ms, post-solution onset there was a main effect of Format (F(1,28) = 37.60; *p* < 0.0001; η_p_^2^ = 0.736), where digit solutions elicited an overall smaller P2 (2.16μV; SE = 0.374) than both LA+ (5.07 μV; SE = 0.475; t(28) = −7.933; *p* < 0.0001) and LA- (4.77 μV; SE = 0.379; t(28) = −7.547; *p* < 0.0001). There was an interaction between Format and Electrode (F(1,50) = 58.62; *p* < 0.0001; η_p_^2^ = 0.677). A distributional analysis revealed a larger effect of Format on Right, Medial, Frontal sites (Format × Hemisphere × Laterality × Anteriority; F(1,28) = 4.02; *p* < 0.01). In this time window, the languages did not differ overall (t(28) = 0.873; *p* = 0.39). There was no main effect of Correctness (F(1,28) = 0.021; *p* = 0.885) or interaction between Format and Correctness (F(1,28) = 2.533; *p* = 0.098).

#### 3.1.4. P300 Time Window (270–470 ms)

This analysis revealed a main effect of Correctness (F(1,28) = 46.072; *p* < 0.0001; η_p_^2^ = 0.122), where correct solutions elicited a more positive brain response (5.56μV; SE = 0.492) than incorrect solutions (3.27 μV; SE = 0.481). There was no main effect of Format (F(1,28) = 1.878; *p* = 0.172) or interaction between Format and Correctness (F(1,28) = 0.844; *p* = 0.441), indicating that the size of the correctness effect did not differ by format.

#### 3.1.5. Post-P300—Early (475–640 ms)

To assess potential effects present immediately after the P300, mean amplitudes were measured between 475–640 ms after solution onset. Analysis of this window revealed a main effect of Format (F(1,28) = 6.73; *p* = 0.004; η_p_^2^ = 0.333), where digits elicited a more negative response overall (3.46 μV; SE = 0.500) compared to LA+ (4.99 μV; SE = 0.529; t(28) = -3.41; *p* = 0.002) and LA− (4.69 μV; SE = 0.386; t(28) = −2.80; *p* = 0.009). Again, the languages did not differ overall (t(28) = 0.633; *p* = 0.532). A main effect of Correctness failed to reach significance (F(1,28) = 1.045; *p* = 0.315), and there was no interaction between Format and Correctness (F(1,28) = 1.00; *p* = 0.379). When splitting the data by native language and second language (i.e., Spanish versus English) there was still no main effect of Language (F(1,28) = 2.77; *p* = 0.107) or interaction between Language and Correctness (F(1,28) = 0.528; *p* = 0.474).

#### 3.1.6. Post-P300—Late (640–900 ms)

At around 640 ms post-solution onset, a later effect seemed to appear in the same direction as the P300 correctness effect. Factorial analysis revealed a main effect of Format (F(1,28) = 3.67; *p* = 0.039; η_p_^2^ = 0.214), where digits again elicited a more negative response overall (3.30 μV; SE = 0.506) compared to LA+ (4.40 μV; SE = 0.661; (t(28) = −2.24; *p* = 0.033) and LA− (4.23 μV; SE = 0.409; t(28) = 0.291; *p* = 0.773). Again, the languages did not differ overall (t(28) = 0.291; *p* = 0.773). There was no main effect of Correctness (F(1,28) = 2.15; *p* = 0.154) or interaction between Format and Correctness (F(1,28) = 0.546; *p* = 0.586).

#### 3.1.7. Experiment 1A Summary of Results

In sum, adults showed comparable response times across formats, but verified digit trials more accurately compared to the spoken word trials. Importantly, bilingual’s performance did not differ when comparing the languages directly (i.e., LA+ versus LA−). All three tasks elicited a P300 ERP correctness effect, in line with previous studies, where a larger P300 was elicited to correct problems compared to incorrect problems. Moreover, there were no significant differences across formats in the size of the effect. However, the ERPs did reveal format effects between digit and spoken word trials before and after the P300, where the response to digit trials was overall more negative than the spoken word trials.

### 3.2. Speeded Multiplication Task—Experiment 1B

#### 3.2.1. Behavior

Factorial analysis of variance was used to analyze accuracy and response times separately, measured from the onset of the solution. Each 2 × 2 ANOVA had two levels of Correctness (correct, incorrect) and two levels of Language (LA+, LA−). Unlike Experiment 1A, there was a significant effect of Correctness on accuracy (F(1,27) = 15.89; *p* < 0.001; η_p_^2^ = 0.371), with more accurate responses for correct (M = 94.05%; SE = 1.04) than incorrect solutions (M = 90.10%; SE = 1.64). Similar to Experiment 1A, there were no main effects of Language (F(1,27) = 1.63; *p* = 0.21) or interaction between Language and Correctness (F(1,27) = 2.16; *p* = 0.15) observed when languages were compared without the digit format.

For response time, only accurate responses were included in the analysis. As in Experiment 1A, there was a significant main effect of Correctness (F(1,27) = 94.64; *p* < 0.001; η_p_^2^ = 0.778), with faster responses to correct (M = 1014.91 ms; SE = 63.51) than incorrect solutions (M = 1212.48 ms; SE = 71.88). There was no main effect of Language (F(1,27) = 0.85; *p* = 0.37) or interaction between Language and Correctness (F(1,27) = 3.14, *p* = 0.09; see Figure 3).

#### 3.2.2. ERPs

All ERPs were measured from the onset of the Arabic digit solution in both languages. Only accurate trials and only participants with a minimum of 20 trials per critical condition were included in analyses (Experiment 1B Average number of trials: LA+ correct—mean: 47, range: 34–54; LA+ incorrect—mean: 46, range: 26–54; LA− correct—mean: 47, range: 29–55; LA− incorrect—mean: 45, range: 21–55). Visual inspection of the grand averaged ERPs revealed sensory components (N1-P2) across the head. The ERPs for correct and incorrect trials began to deviate around 300 ms, with correct solutions eliciting a positive-going deflection, most visibly at occipital electrodes, compared to incorrect solutions (Figure 5). This occurred for both language conditions. The positive deflection had no clear peak and lasted at some electrodes through the end of the epoch. Three separate time windows were selected to assess any mean amplitude differences across formats at the P200, P300, and post-P300 window.

A separate 2 (Correctness: correct, incorrect) × 2 (Language: LA+, LA−) ANOVA was performed on measurements of mean amplitude within the following time windows.

#### 3.2.3. P200 (200–270 ms)

This analysis revealed a main effect of Correctness (F(1,28) = 5.156; *p* = 0.031; η_p_^2^ = 0.155), where correct solutions elicited a smaller P2 (4.433 μV; SE = 0.336) than incorrect solutions- (4.949 μV; SE = 0.422). There was no main effect of Language (F(1,28) = 1.835; *p* = 0.187) or interaction between Language and Correctness (F(1,28) = 0.022; *p* = 0.882).

#### 3.2.4. P300 (340–540 ms)

This analysis revealed a main effect of Correctness (F(1,28) = 25.527; *p* < 0.0001; η_p_^2^ = 0.469), where correct solutions elicited a more positive brain response (3.609 μV; SE = 0.431) than incorrect solutions (2.124 μV; SE = 0.529). Similar to Experiment 1A, there was no main effect of Language (F(1,28) = 0.112; *p* = 0.740) or interaction between Language and Correctness (F(1,28) = 0.191; *p* = 0.665), indicating that the size of the correctness effect did not differ across languages. When splitting the data by native language and second language (i.e., Spanish v. English) there was still no main effect of Language (F(1,28) = 0.70; *p* = 0.41) or interaction between Language and Correctness (F(1,28) = 0.614; *p* = 0.440).

#### 3.2.5. Post-P300 (640–900 ms)

At around 640ms post-solution onset, a later sustained effect seemed continue after the P300. Factorial analysis revealed a main effect of Correctness (F(1,28) = 11.405; *p* = 0.002; η_p_^2^ = 0.287), where correct solutions remained more positive (3.268 μV; SE = 0.520) compared to incorrect solutions (2.175 μV; SE = 0.601). Again, the languages did not differ overall, with no main effect of Language (F(1,28) = 1.171; *p* = 0.289). Additionally, there was no interaction between Language and Correctness (F(1,28) = 0.137; *p* = 0.714).

#### 3.2.6. Experiment 1B Summary of Results

In summary, bilingual adults showed a correctness effect for behavioral measures, where they verified correct solutions faster and more accurately than incorrect solutions overall. However, their performance across languages (LA+ vs. LA−) did not differ significantly. A similar pattern was shown in the ERPs. There was a main effect of correctness in all three time windows that were measured, where correct solutions elicited a more positive response than incorrect solutions. Importantly, this sustained effect did not significantly differ across languages.

### 3.3. Baysian Analyses

To confirm that the languages did not differ based on mean amplitude within the P300 time windows in Experiment 1A and 1B, we also modeled our data using Bayesian analysis. The data were modeled based on Correctness, Language, and the additive effect with and without the interaction between the two. A Watanabe information criteria comparison [66] showed that the models without the interactions best fit the data, and posterior summaries of the model parameters provide Bayesian p-values, which reflect the likelihood of disconfirming a null effect (i.e., zero difference across conditions) with p-values considered significant when the 95% highest posterior density for each parameter did not contain 0. For mean amplitudes within the P300 time window in Experiment 1A, the model revealed a Bayesian p-value of 100.00% for Correctness and a 68.03% Bayesian p-value for Language. The interaction term showed insignificant parameter values, with a Bayesian p-value of 59.85%. For mean amplitudes within the P300 time window in Experiment 1B, the model revealed a Bayesian p-value of 99.62% for Correctness and a 57.98% Bayesian p-value for Language. The interaction term again showed insignificant parameter values, with a Bayesian p-value of 59.17%. In brief, the Bayesian modeling confirms the ANOVAs showing no main effect or interaction with language.

### 3.4. Comparison of Experiment 1A and B

The following analysis are direct comparisons between experiments 1A and 1B for each dependent measure separately. Only LA+ and LA− were compared across experiments since Experiment 1B did not have digit trials. Separate repeated measures ANOVA with Language (LA+, LA−) and Correctness (correct, incorrect) as within-subject factors and Experiment (A, B) as a between-subject factor were used for each dependent measure, except for ERP latency. Analyses for onset and peak latency for the ERP effects were measured on difference waves (incorrect minus correct trials) using an ANOVA with Language (LA+, LA−) as the within-subject factor and Experiment (A, B) as a between-subject factor. All participants in Experiment 1A (*n* = 29) were compared to participants in Experiment 1B (*n* = 28). Note that although 12 individuals participated in both experiments, we treated Experiment (A, B) as a between-subject factor for a more conservative metric. The analyses were also conducted without these 12 participants, and on these 12 participants separately; the pattern of results did not change for any measures.

#### 3.4.1. Behavior

Overall accuracy did not differ between Experiment 1A (mean: 89.48%; SE: 1.47) and Experiment 1B (mean: 92.07%; SE: 1.50; F(1,55) = 1.510; *p* = 0.224). There was a main effect of Correctness (F(1,55) = 11.80; *p* = 0.001; η_p_^2^ = 0.177), but no interaction between Correctness and Experiment (F(1,55) = 2.54; *p* = 0.117). There was also no main effect of Language (F(1,55) = 0.093; *p* = 0.762) or other interactions. Response Time also did not differ between Experiment 1A (mean: 960.041 ms; SE: 65.97 ms) and Experiment 1B (mean: 1113.69 ms; SE: 67.14 ms; F(1,55) = 2.66; *p* = 0.108). There was a main effect of Correctness (F(1,55) = 158.99; *p* < 0.0001; η_p_^2^ = 0.743), but no interaction between Correctness and Experiment (F(1,55) = 2.95; *p* = 0.091). There was also no main effect of Language (F(1,55) = 0.311; *p* = 0.580) or other interactions. In addition, there were no measurable effects of language for either Experiment 1A (F(1,28) = 1.22; *p* = 0.279) or 1B (F(1,27) = 0.03; *p* = 0.86) when comparing response times between English and Spanish, indicating that word length differences did not affect processing speed.

#### 3.4.2. P200 (200–270 ms)

There were no effects of Language (F(1,55) = 0.003; *p* = 0.95), Correctness (F(1,55) = 3.815; *p* = 0.056), or Experiment (F(1,55) = 0.192; *p* = 0.663), and no interactions between these factors (*p*’s > 0.10) on mean amplitude at the P200. This reflects that the effect at the P200, which is possibly driven by the switch between formats, was not affected by speeding up that task.

#### 3.4.3. P300 (Variable Time Window)

Latency: Using the peak latency measurement tool in ERPLab on the difference waves (incorrect minus correct), we measured the latency of the most positive peak within a broad window (200–500 ms). The toolbox defines a peak as (1) being the largest positive value of all data samples in the entire window and (2) being larger than the average of 3 sample points on both the left and right sides of the peak. The output allowed us to determine when the effect was maximal across conditions, collapsed across all channels and participants.

Overall onset latency for the correctness effect was significantly earlier by 59.78 ms for Experiment 1A (mean: 299.511 ms; SE = 5.7 ms) than Experiment 1B (mean: 359.29 ms; SE: 5.96 ms) (F(1,55) = 53.178; *p* < 0.0001; η_p_^2^ = 0.492). This delay is consistent with prior studies where faster presentation rates led to a delay in the onset of ERP effects (Dickson et al., 2018). This effect provides evidence that the speeded task increased overall processing demands compared to Experiment 1A. There was no main effect of Language (F(1,55) = 0.013; *p* = 0.911) or interaction between Language and Experiment (F(1,55) = 0.970; *p* = 0.329).

Similarly for peak latency, a repeated measures ANOVA over a common extended window (300–880 ms) showed a significantly earlier peak for the correctness effect in Experiment 1A (498.969 ms; SE = 20.644) than Experiment 1B (562.412; SE = 21.010) (F(1,55) = 4.639; *p* = 0.036; η_p_^2^ = 0.695). There was no main effect of Language (F(1,55) = 1.392; *p* = 0.243) or interactions between these factors (F(1,55) = 0.354; *p* = 0.554).

Mean Amplitude: Given the significant latency difference across experiments, two time windows of identical length were used to measure mean amplitude of the P300 (Experiment 1A: 270–470 ms; Experiment 1B: 340–540 ms). There was a main effect of Experiment (F(1,55) = 7.301; *p* = 0.009; η_p_^2^ = 0.117), where the response to Experiment 1A was overall more positive (4.618 μV; SE = 0.454) than Experiment 1B (2.866 μV; SE = 0.462). As expected, there was a main effect of Correctness (F(1,55) = 56.634; *p* < 0.0001; η_p_^2^ = 0.507) with larger P300 amplitude for correct (4.652 μV; SE: 0.328) than incorrect trials (2.832 μV; SE: 0.363), and no main effect of Language (F(1,55) = 0.567; *p* = 0.455). There were no interactions between any of the factors (*p*’s > 0.15), indicating that the presentation rate did modulate the effects of Language or Correctness. See Figure 6 for comparison of P300 Correctness effect topographies across Experiment 1A and 1B.

#### 3.4.4. Post-P300 (640–900 ms)

There was a main effect of Experiment (F(1,55) = 5.035; *p* = 0.029; η_p_^2^ = 0.084), where the response to solutions in Experiment 1A was overall more positive (4.316 μV; SE = 0.498) than solutions in Experiment 1B (2.721 μV; SE = 0.507). As expected, there was a main effect of Correctness (F(1,55) = 10.608; *p* = 0.002; η_p_^2^ = 0.162) with more positive amplitude for correct (3.962 μV; SE = 0.369) than incorrect trials (3.075 μV; SE = 0.391), and no main effect of Language (F(1,55) = 0.680; *p* = 0.413). There were no interactions between any of the factors (*p*’s > 0.40), indicating that the presentation rate did modulate the effects of Language and Correctness.

#### 3.4.5. Comparison of 1A and 1B Summary of Results

Bilingual adults showed a correctness effect in both accuracy and response time overall, but performance did not differ across experiments (1A versus 1B). At the P200, there were no significant differences across experiments. The P300 correctness effect in Experiment 1A occurred earlier (~60 ms) and was overall more positive than Experiment 1B. After the P300, there was a main effect of correctness, and the response in Experiment 1A remained significantly more positive than Experiment 1B. There were no interactions by language in any of these measures across experiments.

### 3.5. Word–Picture Verification (WPV)

#### 3.5.1. Behavior

Behavioral measures for the WPV task were measured based on participants’ first (L1) and second languages (L2). All participants reported an L1 of Spanish, while 11 out of the 29 participants reported Spanish as their LA+. Separate factorial analyses of variance were used for accuracy and response time measured from the onset of the picture with two levels of Congruency (congruent, incongruent) and two levels of Language (L1, L2). For accuracy, there was a main effect of Language (F(1,26) = 6.906; *p* = 0.014; η_p_^2^ = 0.210) where participants were overall more accurate in L2 (97.61%; SE = 0.386) compared to L1 (95.97%; SE = 0.598). This likely reflects the greater use of L2 English in daily life (82% average daily use of English). There was also a main effect of Congruency (F(1,26) = 6.459; *p* = 0.017; η_p_^2^ = 0.199), where participants were more accurate in answering incongruent (97.65; SE = 0.439) compared to congruent (95.92%; SE = 0.591) word–picture pairs. Despite the high accuracy for both conditions, this may reflect a slight speed–accuracy tradeoff prioritizing speed over accuracy. This more likely indicates that rejecting an incorrect label was easier than confirming a specific correct label for a picture.

However there was no interaction between Language and Congruency (F(1,26) = 0.07; *p* = 0.794). Response times were significantly faster for congruent (712.30 ms; SE = 32.69) than incongruent (751.09 ms; SE = 29.93) trials (main effect of Congruency; F(1,26) = 11.96; *p* = 0.002; η_p_^2^ = 0.315). There was no main effect of Language (F(1,26) = 0.23; *p* = 0.630) or interaction between Language and Congruency (F(1,26) = 0.80; *p* = 0.379) for response time (see Figure 7).

#### 3.5.2. N400 (280 ms–480 ms)

All ERPs were measured from the onset of the line drawings for the word–picture verification task in both languages. ERPs were averaged separately for congruent and incongruent pictures; only trials that subjects judged accurately were included in analyses. Visual inspection of the grand average ERPs revealed typical sensory components, N1-P2, across the head in response to the line drawings for all conditions. Around 200 ms after solution onset, a broadly distributed negative-going deflection is observed with a larger amplitude for incongruent than congruent pictures—an N400 congruency effect (Figure 8). This effect was present for both languages.

To determine the timing of the N400 congruency effect across languages, averaged responses to congruent pictures were subtracted from incongruent pictures to create difference waves. Using the peak latency measurement tool in ERPLab (as above), a broad window (200–500 ms) was used to determine when the effects reached maximum amplitude collapsed across channels and across participants. Measuring from the onset of the solution, the effect reached peak amplitude at 378 ms (SE = 6.09) for L1 and 376 ms (SE = 7.16 ms) for L2. The timing of the N400 effect across languages did not significantly differ (t(26) = −0.293, *p* = 0.772). Therefore, a single time window (280 ms–480 ms) was used in both languages for measurements of mean amplitude.

Mean amplitude was analyzed using a 2 (Congruency: congruent, incongruent) × 2 (Language: L1, L2) × 26 (Electrode) ANOVA. This revealed a main effect of Congruency (F(1,26) = 140.24; *p* < 0.0001; η_p_^2^ = 0.844), where incongruent pictures elicited a more negative brain response (0.242 μv; SE = 0.704) than congruent pictures (4.26 μv; SE = 0.804). A significant interaction between Congruency and Electrode (F(1,26) = 20.13; *p* < 0.0001; η_p_^2^ = 0.436), suggested a typical N400 distribution. A distributional analysis (see Experiment 1) replacing Electrodes with 2 levels of Laterality, 2 levels of Hemisphere, and 4 levels of Anteriority confirmed the presence of a larger effect over medio-central sites (Congruency × Laterality × Anteriority; F(1,26) = 6.71; *p* = 0.002). There was no main effect of Language (F(1,26) = 0.005; *p* = 0.942) or interaction between Language and Congruency (F(1,26) = 0.913; *p* = 0.348).

#### 3.5.3. Post-N400 (500–900 ms)

At around 640 ms post-solution onset, a later sustained effect seemed to continue after the N400. Factorial analysis revealed a main effect of Congruency (F(1,28) = 18.632; *p* < 0.0001; η_p_^2^ = 0.417), where congruent trials elicited a more positive response (6.77 μV; SE = 0.637) than incongruent trials (5.329 μV; SE = 0.620). There was no main effect of Language (F(1,28) = 0.055; *p* = 0.816) or interaction between Language and Congruency (F(1,28) = 0.200; *p* = 0.658).

#### 3.5.4. WPV Task Summary of Results

Participants were overall more accurate verifying word–picture pairs in L2 compared to L1. This likely reflects the greater use of L2 English in daily life. Despite the high accuracy for both conditions, responses to incongruent trials were slower and more accurate than correct trials, which may reflect a speed–accuracy tradeoff prioritizing speed over accuracy. This may also indicate that, given their larger vocabulary, adults may have experienced more uncertainty about the correct picture labels, whereas rejecting an incorrect label was more obvious. There was no difference in the congruency effect across languages. In the ERPs, this task elicited an N400 congruency effect that was morphologically distinct from the P300 correctness effect elicited to the multiplication tasks in Experiment 1A and 1B. This ERP effect did not significantly differ across languages.

## 4. Discussion

The current study is built off of a small number of experiments investigating the neural time course for multiplication verification in the bilingual brain [4,6,14]. The primary goal was to resolve discrepancies in methodology across these studies, and the monolingual literature, to better understand how bilinguals process multiplication in each of their languages. More specifically, this study tested the hypothesis that the adult bilingual brain should show differences in processing either in the type (based on TCM) or size (based on ECM) of the brain’s response when verifying multiplication facts in the language the facts were originally learned (LA+) and the other language (LA−). We designed the study following recent discoveries in the ERP arithmetic literature in monolinguals, such as format effects and delayed response effects. We also applied methodological controls from the bilingual literature by removing linguistic confounds at the solution and controlling for participant language experience. In this way we tested the hypothesis that bilinguals access multiplication facts preferentially in the language of learning arithmetic without potential linguistic or task-specific confounds.

We measured the ERP correctness effect for simple multiplication problems with immediate behavioral responses from solution onset. The operands were presented either as spoken words in LA+ or LA−, and trials were presented with or without inter-stimulus intervals (ISIs) to manipulate presentation rate. In all cases the solution was a digit. This allowed us to measure the effects of fact retrieval based on the format of the preceding operands while eliminating lexical effects at the solution itself. Finally, we compared the findings from the spoken multiplication task to a language task known to elicit an N400 [44], as well as trials presented as all digits to measure a P300 correctness effect. This allowed us to address the recent reinterpretation of the ERP correctness effect [15,20,21,22,44] and determine if bilinguals elicit an N400 or P300 in each of their languages.

In brief, the ERP results revealed a P300 to correct solutions in all three formats (LA+, LA−, and digits), with larger positive amplitude for correct than incorrect solutions, consistent with the monolingual adult literature [15,20,21,22,23,24,25,44]. This effect was clearly distinct from the N400 effect observed in the word–picture matching task. Critically, the P300 correctness effect was not different between LA+ and LA− in timing, amplitude, or distribution, contrary to the original bilingual arithmetic ERP findings [4,14]. Even when performing the task with speeded trials, there was no difference between languages in these robust ERP effects. The N400 effect observed in the word–picture task was also indistinguishable between languages, reflecting the equivalent language proficiency of our bilingual sample. Similarly, the behavioral findings revealed no measurable language bias in response time or accuracy in judging the correctness of the solutions. We discuss the implications of these findings below.

### 4.1. N400 versus P300 Interpretation of the Correctness Effect

As discussed in the introduction, this study was conducted in the midst of a reevaluation of the original ERP correctness effect. Studies using multiplication problems presented as digits originally interpreted the correctness effect as a modulation of the N400, thought to index access to semantic memory [4,23,24,25]. Subsequent studies demonstrated that the ERP effect was actually a modulation of the P300, indicating that adults treat correct solutions as targets in overlearned math facts [15,20,21,22]. Based on this reinterpretation, we predicted that bilingual adults would also show a robust P300 correctness effect for all-digit trials, which was confirmed as a larger positive amplitude for correct than incorrect solutions.

The prediction for trials with spoken number words was less clear. On the one hand, Salillas and Wicha [4] reported N400 correctness effects for multiplication problems presented as written words in bilingual adults. On the other hand, monolingual adults who performed the same task used herein showed a P300 correctness effect at the solution, with no differences in the timing or amplitude of the effect between digit and spoken number word trials [21]. If spoken word trials engage more lexico-semantic processes, then bilinguals might have exhibited an N400, at least in the less automatic LA-. Instead, bilinguals showed a robust P300 response to solutions in spoken word trials. This effect was clearly distinct from the N400 effect elicited in the word–picture verification task (Experiment 2) and was indistinguishable statistically from the P300 for digit trials. Like monolinguals, bilinguals treated the solutions as targets in overlearned problems and did not process the problems for meaning, categorizing the spoken word trials as targets (correct) and non-targets (incorrect) just like the digit trials (We observed other format effects at the solution before and after the P300 that were driven by differences between digits and words, but not between LA+ and LA−. We discuss these effects in the results section for completeness but focus the discussion section on the analysis that are most relevant to the hypothesis tested herein).

Bilinguals also showed the same behavioral pattern as monolinguals, with more accurate (but not faster) overall responses for solutions in digit trials than spoken number word trials (Experiment 1A). Notably, TCM might have predicted the opposite effect with faster times for spoken number words than digits [2,17]. ECM would account for this difference as weaker access to representations of arithmetic facts in the word form compared to the digit form [16,67]. However, the lack of a P300 format effect at the solution reveals that the categorization of solutions as correct or incorrect is not affected by the format of the preceding operands. Therefore, the effect of format on accuracy may be in response generation rather than retrieval from memory.

### 4.2. Language of Learning Effects

Most relevant to the goal of this study, the P300 effect at the solutions was indistinguishable between LA+ and LA− trials. There were also no effects of language on either response time or accuracy. Speeding up the task caused a latency delay for the P300 and slower response times, but there was still no evidence of a language of learning effect in either the ERPs or behavioral measures. This was again in contrast to Salillas and Wicha [4], where adult bilinguals showed a larger N400 effect for LA+ than LA− trials.

Several important methodological differences might explain the contrast between the original bilingual findings and the current study.

First, we must address the difference between the N400 effect observed in Salillas and Wicha [4] and Martinez-Lincoln et al. [14] and the P300 observed herein to number word trials. Salillas and Wicha [4] used written words for both operands and the solution itself. The current study adapted the Cerda et al. [6] paradigm where the first two operands were presented as spoken number words in English or Spanish on separate trials. The solution was always presented as a digit (e.g., “two” “three” 6). In this way, effects at the solution would reflect access to the multiplication facts based on the preceding spoken operands, and not lexical differences at the solution itself. It is possible that using written words, or words in general, at the solution encouraged lexico-semantic processing, in turn eliciting a modulation of the N400. However, Dickson et al. [15] observed an N400 to digit solutions preceded by spoken words, so neither of these conditions seem necessary for eliciting an N400. What seems more critical for generating an N400 is the use of delayed responses.

Dickson et al. [15] and Salillas and Wicha [4] imposed a delay where participants had to hold their response for one second after the solution appeared before responding. The motivation for this was to avoid motor contamination of the ERPs to the solution, but this delay seems to encourage the engagement of semantic level processing. Using similar stimuli but changing the task to an immediate response upon seeing the solution, adults appear to process the solutions more superficially, with no evidence of a semantic level N400 modulation [21]. Instead, the correct solutions elicit a P300 as if they were targets in a target detection task or easier items to categorize in a categorization task. It is possible then that the immediate response itself changes how the problems were processed.

With that said, what is the significance of this difference? On the one hand, we show that bilinguals can verify multiplication problems equivalently in both LA+ and LA−. This indicates that LA+ does not have a stronghold on better processing for arithmetic in the bilingual brain. On the other hand, the differences observed in Salillas and Wicha [4] and Martinez-Lincoln et al. [14] reveal that LA+ can show preferential processing under certain circumstances. The N400 observed in these earlier studies indexes access to semantic memory [35,36,68].

In contrast, the P300 observed herein indexes more superficial memory processes involved in the categorization of these overlearned math facts [15,20,21,22,44]. Therefore, perhaps differences do exist in the lexico-semantic representation of math facts in LA+ and LA− that are only captured by a task that elicits an N400. We caution this interpretation given that other methodological choices could account for this effect. We discuss below how the current study helps determine whether these differences are due to the representation of math facts themselves, or the other task-related confounds.

It is possible that the LA+ effect observed in Salillas and Wicha [4] was driven by the use of written word stimuli, especially at the solution. The difference between LA+ and LA− might reflect reading ability or reading frequency differences across the languages, with less efficient processing in the LA−. It is possible that these language differences were driven by weaker recognition of the math facts in the written form or differences in reading fluency in a population that reads primarily in English. By using spoken number words in the current study, we eliminated this potential confound. Interestingly, even written number word processing can become equivalent across LA+ and LA− with increased experience using arithmetic facts in the weaker language [14]. It is possible that the limited experience using the written word form might elicit language of learning effects.

Perhaps an even more compelling explanation for the difference in language of learning effects is the mixed versus blocked presentation of the trials. While in this study problems were blocked by language, Salillas and Wicha [4] presented the stimuli mixed with consecutive trials appearing at random, with alternating languages. Research has shown that switching between languages on linguistic tasks causes asymmetric interference effects in a dominant versus non-dominant language [29,30,31,32]. In turn, the task used by Salillas and Wicha [4] may have elicited differences across languages, not because of differences in access to the multiplication facts, but because of asymmetric language switching costs.

In fact, McClain and Huang [33] reported that language switching alone can lead to language differences for arithmetic. They presented Spanish–English bilinguals with simple addition and multiplication problems presented in either alternating language blocks or in two separate language sessions. When participants performed the arithmetic task in separate language sessions there were no differences in response time between the preferred or non-preferred language. Language differences only arose when subjects were required to work in both languages within a single session (i.e., language switching). An asymmetric switching effect has also been observed for number word reading in bilingual adults [34].

One limitation of our study design is that participants may be transcoding (or translating) consciously or not from the spoken words to the digit format. That is, it is possible that upon hearing the spoken number words participants imagine the digit, such that on arriving at the digit solutions there is little difference in processing across conditions. This could potentially explain the lack of differences between languages, assuming that bilinguals are equally able to transcode from each spoken language to the digit format. Although this is possible, it is unlikely to explain the current results given that format effects did occur before and after the P300. Specifically, the effect of Format at the P2 most likely reflects the cross-modal nature of the auditory multiplication trials. Deviant stimuli in classic oddball paradigms elicit larger P2 components [69], whereas repetition of a single stimulus results in reduced P2 amplitudes [70]. This might suggest that participants were still perceiving a switch across modalities at the presentation of the solution. The differences at the P2 and after the P300 in overall amplitude occurred at the onset of solutions that were identical across conditions (i.e., digit). The only difference between conditions was the format of the preceding operands. Therefore, format effects at the solution had to be driven by differential processing of the preceding operands.

Lastly, there is evidence that linguistic factors play a causal role in the language bias for arithmetic [10,13]. Our population of bilinguals were native Spanish speakers who learned English early in life and were equally proficient in both English and Spanish at the time of testing. We confirmed their equivalent proficiency in multiple ways, based on objective standardized measures of language ability and the N400 effects on the language task. In this way, we could measure differences in access to math facts across languages without the confound of general language ability. Many prior studies have included bilinguals with a range of language backgrounds (late learners of a second language and people with dominant proficiency in one of their languages), most inferring language proficiency from self-report. Therefore, it is possible that the absence of a difference across languages in the performance or ERP measures reflects our sample’s balanced language proficiency.

## 5. Conclusions

In summary, our study revealed that fluent bilingual adults can process multiplication facts equivalently in both languages, despite having learned math in one language (LA+). By carefully controlling for linguistic factors, both in the stimuli and the included participants, and rigorously considering methodological confounds, such as delayed responding and language switching, we were able to test for differences in representation across the languages. Our findings suggest that previously reported differences in performance and brain activity across LA+ and LA− may be caused by methodological choices, such as switching trials between languages, rather than differences in memory encoding or retrieval. This is in line with a recent meta-analysis study showing that language factors can predict differences in arithmetic performance across languages [13]. 

Importantly, we have only tested for language effects during the verification of multiplication facts. Producing the solution may show different results, similar to differences in memory more generally between recognition and recall [71,72,73,74]. Nevertheless, our results can help refine models of arithmetic, like ECM and TCM, and help generate new hypotheses to be tested within these frameworks. For example, the presence of an N400 versus a P300 implies different levels of cognitive processing. In turn, the representation of math facts in these models must also account for the ability to access this information at superficial pattern recognition and lexico-semantic levels. In addition, the absence of an LA+ advantage supports models like ECM that allow for experience to change the strength of connections between representations. In brief, our results push the field toward a more refined understanding of the source of the language of learning effect and take a more holistic perspective of bilingualism when studying math cognition in the bilingual brain.

To be clear, what is critical here is not that bilinguals show no differences in processing multiplication facts across their languages, because many studies have shown that they can, as mentioned above. Language differences can have real-world implications on the efficiency and accuracy of performing simple arithmetic in daily life [5,75,76,77]. The impact of bilingualism on math cognition continues to be an important avenue of investigation without a doubt. Based on our findings, we predict that bilinguals who are less fluent in one language, such as late learners of a language or children entering the school system in a weaker language, should show less efficient arithmetic verification in the LA−, especially in more demanding tasks. In brief, our findings suggest that linguistic and methodological differences themselves may be the cause of these processing differences across languages. With this understanding, the field can move away from models that emphasize the importance of the language of early learning of math facts and toward a more holistic understanding of the effects of language on arithmetic processes in the bilingual brain.

## Figures and Tables

**Figure 1 brainsci-12-00532-f001:**
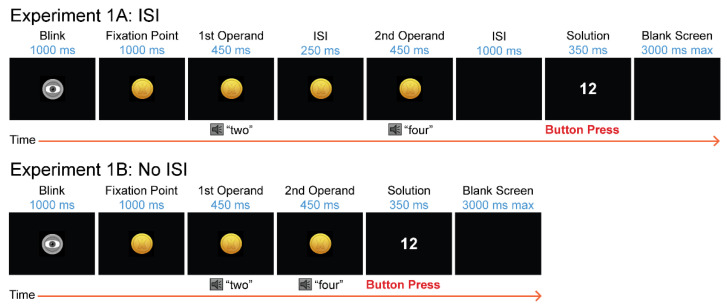
Stimulus timing. An example multiplication trial from the English task in Experiment 1A (top) shows the trial structure from left to right over time in milliseconds. First and second operands were presented as spoken number words. The trial structure was identical for the Digit and Spanish multiplication tasks in Experiment 1A with the operands presented as Arabic digits or spoken number words presented in Spanish. Participants were asked to respond from the onset of the visual solution. An example trial from the English task in Experiment 1B (bottom) shows a trial structure similar to Experiment 1A with the ISIs removed. This speeded trial structure was identical for the Spanish multiplication task in Experiment 1B.

**Figure 2 brainsci-12-00532-f002:**
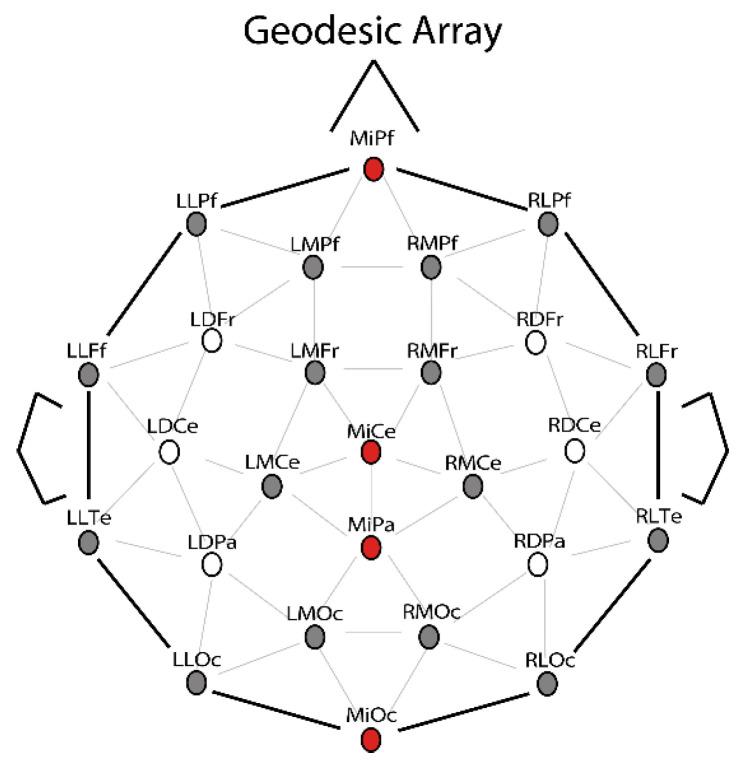
Layout of electrode channels used in all recordings, oriented with the front of the head at the top of the figure. Grey circles indicate electrodes utilized in distributional analyses. Red circles indicate representative midline electrodes used across all subsequent figures for visualization of ERP effects.

**Figure 3 brainsci-12-00532-f003:**
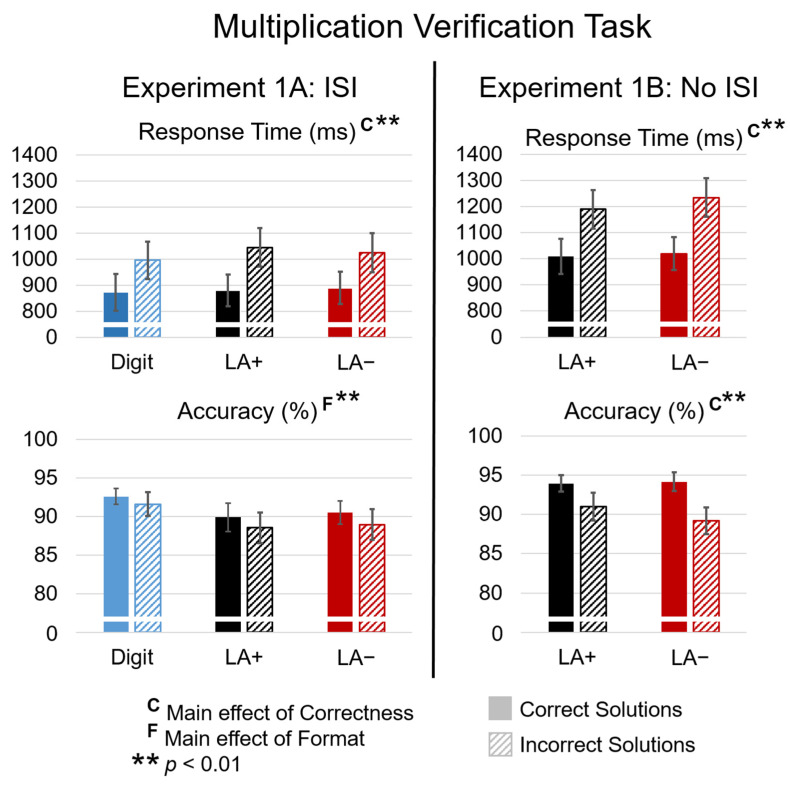
Behavioral results for the multiplication tasks in Experiment 1A (*n* = 29) and Experiment 1B (*n* = 28). The blue bars represent accuracy and response times for the Digit task (only in Experiment 1A). The language of learning(LA+) and the other language (LA−) are depicted for both experiments in the black and red bars, respectively. Response time measurements include only trials that participants judged correctly.

**Figure 4 brainsci-12-00532-f004:**
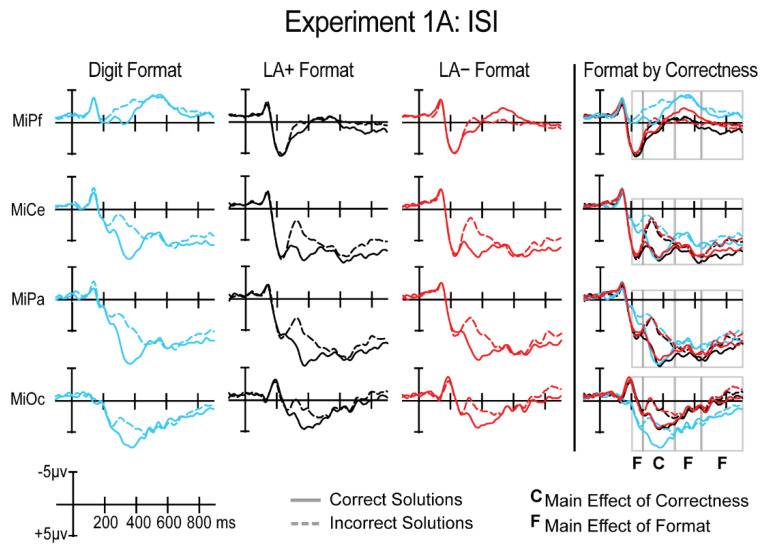
Grand averaged ERPs time-locked to Arabic digit solution onset for multiplication problems presented as digits (blue), spoken number words in LA+ (black), and spoken number words in LA— (red). Representative midline electrodes are shown. Responses to correct solutions are represented as solid lines and responses to incorrect solutions are represented as dotted lines across all formats. Traces on the far right depict the overlap of all three formats, highlighting consecutive time windows used for analysis in the grey boxes (*n* = 29). Significant effects are indicated below the windows; *p* < 0.01.

**Figure 5 brainsci-12-00532-f005:**
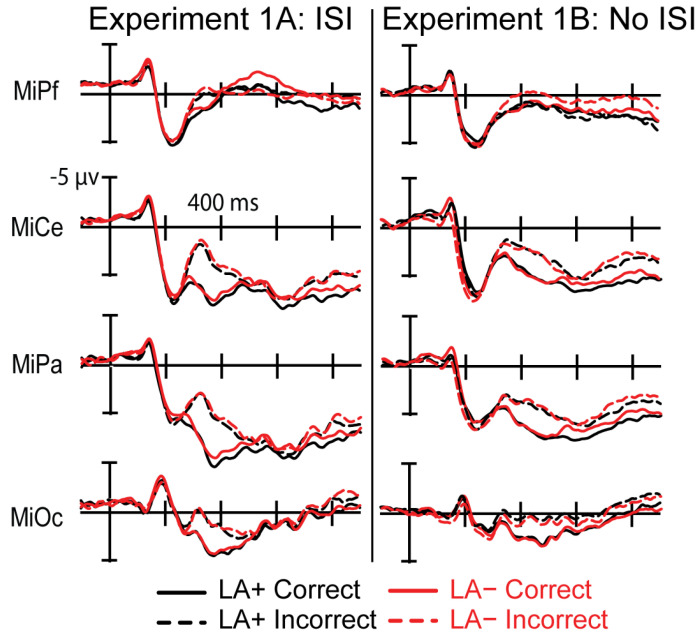
Grand averaged ERPs time-locked to Arabic digit solution onset for multiplication problems presented as spoken number words in LA+ (black) and spoken number words in LA− (red) with and without interstimulus intervals (ISIs) during the presentation of the problem. Representative midline electrodes are shown. Responses to correct solutions are represented as solid lines and responses to incorrect solutions are represented as dotted lines.

**Figure 6 brainsci-12-00532-f006:**
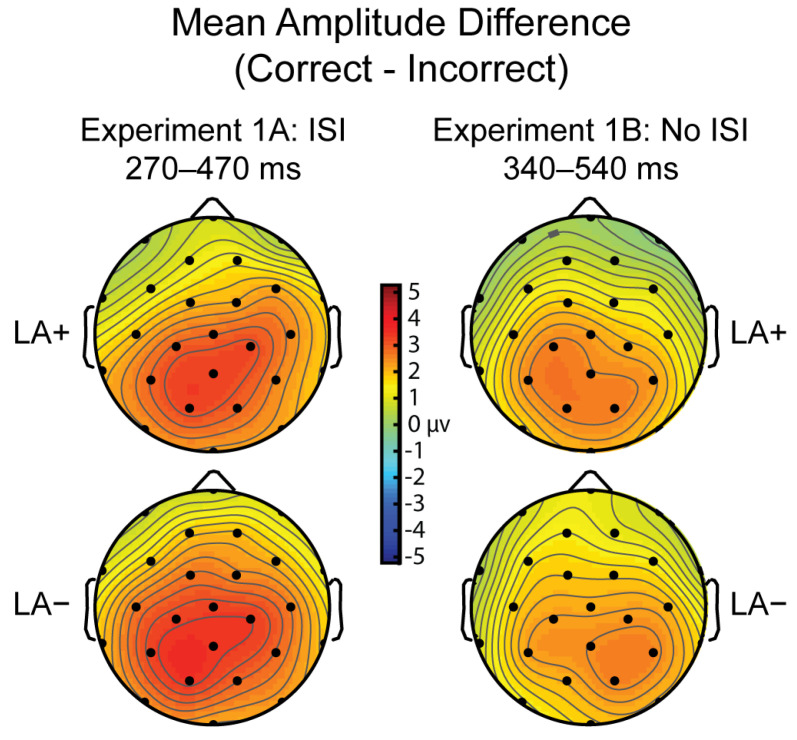
Topographic (isovoltage) maps of the correctness effect (correct–incorrect) for each language between 270 and 470 ms after solution onset for Experiment 1A (left) and between 340 and 540 ms after solution onset for Experiment 1B (right).

**Figure 7 brainsci-12-00532-f007:**
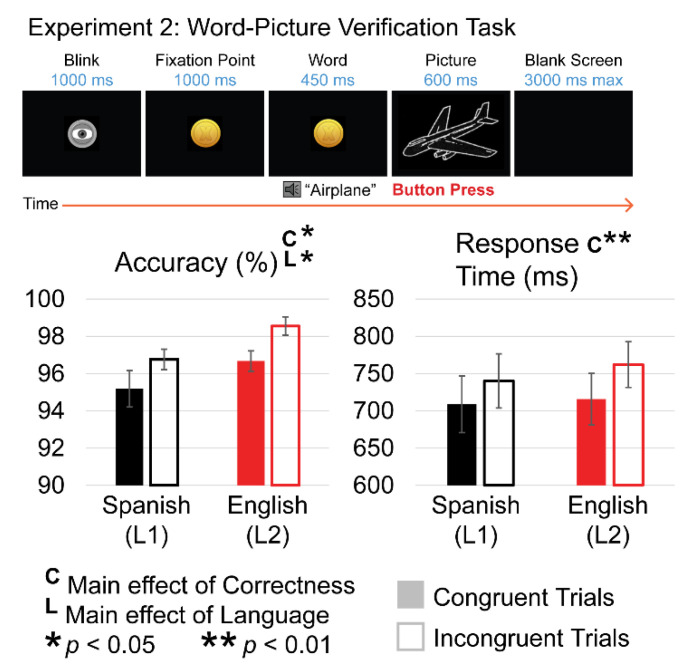
Word–Picture Verification task. An example trial in English showing the trial structure from left to right over time in ms is depicted on the top. Trial structure was identical for the Spanish task. Participants were asked to respond from the onset of the line drawing. Behavioral results for the Word–Picture Verification task are depicted in the bar graphs. Percent accuracy and response time in milliseconds measured from solution onset (*n* = 27). Dark bars indicate trials with congruent picture–word pairs and striped bars indicate trials ending with incongruent picture–word pairs. Response time measurements include only trials that participants judged correctly. Significant effects are indicated above the bar graphs; C—Main effect of Correctness; F—Main effect of Format.

**Figure 8 brainsci-12-00532-f008:**
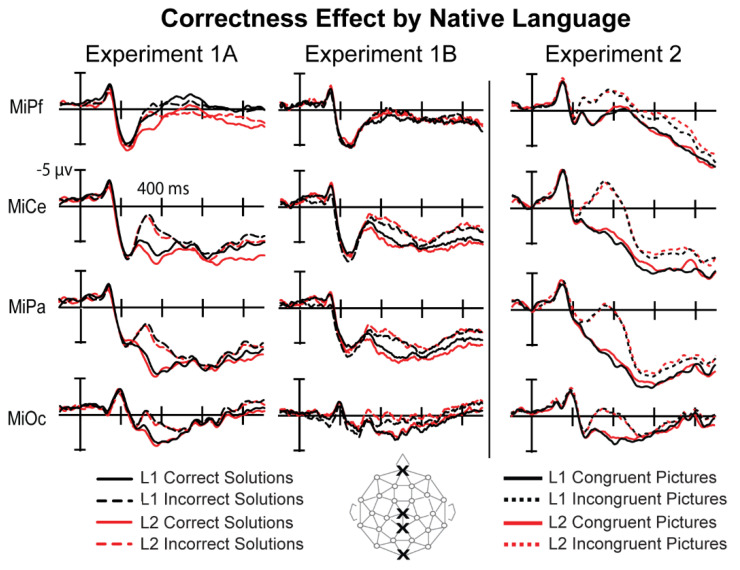
ERPs by native language. Grand average ERPs across midline electrodes time-locked to solution onset for Experiments 1A and 1B (left and center, respectively) and picture onset for Experiment 2 (right). Correct/congruent (solid lines) and incorrect/incongruent (dotted lines) items were preceded by spoken operands (Experiments 1A and 1B) or nouns (Experiment 2) in L1 (black) and L2 (red). It should be noted that L1 and L2 do not necessarily correspond to LA+ and LA− in all individuals. Robust P300 correctness effects are seen for both languages for Experiments 1A and 1B, whereas N400 congruency effects are seen for both languages for Experiment 2. Importantly, there were no differences in the size of the effect across languages for any task.

**Table 1 brainsci-12-00532-t001:** Participant performance on language measures ^1^.

Test	English Standard Score	Spanish Standard Score
Experiment 1A (*n* = 29)		
Picture Vocabulary	87.69 (SD = 10.86)	87.31 (SD = 10.12)
Oral Comprehension	97.69 (SD = 6.05)	97.37 (SD = 9.21)
Incomplete Words	93.10 (SD = 15.48)	88.89 (SD = 8.05)
Experiment 1B (*n* = 28)		
Picture Vocabulary	84.89 (SD = 11.12)	89.85 (SD = 11.25)
Oral Comprehension	95.07 (SD = 7.82)	97.68 (SD = 10.28)
Incomplete Words	87.68 (SD = 18.80)	91.75 (SD = 8.95)

^1^ All mean Standard Scores reflect age-based norms from monolingual English and monolingual Spanish speakers, where 100 is average and 15 points reflects 1 standard deviation outside the norm.

## Data Availability

The data presented in this study are available on request from the corresponding author. The data are not publicly available because the data is part of a larger dataset that continues to be used for investigation.

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
