# Peer review of "Reevaluating the Language of Learning Advantage in Bilingual Arithmetic: An ERP Study on Spoken Multiplication Verification"

_brainsci, 2022, doi:10.3390/brainsci12050532_

Round 1

Reviewer 1 Report

The authors have successfully met the issue of number word length in Spanish and English.

Reviewer 2 Report

Dear authors,

You did an excellent job.

Congratulations.

This manuscript is a resubmission of an earlier submission. The following is a list of the peer review reports and author responses from that submission.

Round 1

Reviewer 1 Report

The paper is structured well and contributes to the literature.

Reviewer 2 Report

The paper aims to address the question whether, in highly proficient bilinguals who have acquired both of their languages early in life, multiplication fact yield similar behavioral and neuronal responses for both of bilinguals languages. The study reports 2 EEG experiments  during multiplication verification using Arabic digits as targets. Operands are either also Arabic digits (XP1) or spoken number words in both of bilinguals languages (XP1 and XP2). In an additional EEG experiment participants have to decide whether visually presented objects match with preceding words presented in both languages. All experiments are follow-ups of previous similar studies, containing certain methodological variations.

These overall questions are timely and interesting. Moreover, it is important to re-evaluate previous studies, and the fact that the authors conducted several experiments could potentially foster the robustness of results. However, I have several major concerns with central theoretical and methodological aspects of the study, which I outline in detail below.

GENERAL COMMENT

It is not clear whether this paper has a methodological aim (i.e. to shed light on the discrepancies between the three previous publications) or a theoretical aim (i.e. find if arithmetic representations have language preferential or exclusive accesses in highly proficient early bilinguals). Because on one side, it states the primary goal is methodological, but then on the other side the paper concludes with strong claims about bilingualism. If the goal is methodological then it would not need such an in-depth theoretical frame and strong predictions while if it is theoretical, then it would not need Experiment 1A.

The ERP analyses also crucially lack clarity and consistency, for example concerning the electrode choices or electrodes used in the ANOVA’s.

In addition, the paper implies the differences are explained by methodological reasons, while the populations between the different papers (including the present one) differ in terms of age of acquisition of the respective languages. In that regard, the methods encompasses also a critical confounder between L1 and LA+, which is the same language for half of the sample (i.e. Spanish), but a different one for the other half (Spanish and English). When averaged this mix might lead to a null effect and/or conflate potential differences for both populations. Indeed, it would be extremely interesting and important to compare the results obtained for participants, as the equal results for both languages might especially stem from the group with English as LA+. If these also apply to the subgroup with Spanish as L1 and LA+, this would have further implications for the interpretation of the results. Moreover, the observed performance advantage for English in the word picture verification task is very difficult to understand in the latter subgroup (if it would confirm) and would need to be and explained and discussed. Finally, the paper is unclear and thus misleading at different critical points such as the aims, the analyses and the conclusion.

INTRO

  • Lines 80-81: I would suggest repeating here the introduced notion of preferential and exclusive accessintroduced at line 30.
  • Lines 120 to 145: Besides the format and blocked/randomized item’s language, the three studies differ alsoin terms of the samples: the samples have different AoA, language use, and language exposure. For examplein Salillas, the adults participants were exposed to English at on average 7.1 y.o., in Cerda L2 AoA is 1.8o., in Martinez-Lincoln at 9.07.

METHODS

Generally, it seems that only Ex1B is relevant for the investigated research question. The necessity of Ex1A and Ex2 is not very clear and generally, the motivation for conducting these two experiments is not clearly articulated.

On a methodological level, it is difficult to make a comparison between Experiment 1 and Experiment 2 given that one of the most important factors, language, is defined differently. While Experiment 1 compares LA+ (Language of mathematical acquisition independently from L1 or L2), Experiment 2 compares L1 and L2 (independently from LA+ and LA-). In that way, it is not only difficult to compare both experiment’s results but also to compare and distinguish the role of the first language of acquisition (L1, L2) vs. the language of consolidation (LA+, LA-) in the result’s interpretation.

Another concern is that the results are simply inconclusive to enable the authors to answer their research questions. Their main interpretation is based on the absence of effect (e.g., no differences between LA+ and LA- and no interaction). However, absence of evidence does not mean evidence of absence, and generally null findings are notoriously difficult to interpret. That being said, I do believe it is only beneficial for science if null findings are reported. Nevertheless, in the current case, I do am not fully convinced it will be easy to do, especially when it is only one study with a rather small sample against all previous literature, and there is also no control condition to compare the null effects to. To begin with, classical statistics can not handle and does warren interpretation of null findings. If the authors would like to interpret the lack of effects, they should resort to another statistical framework, such as the Bayesian, where interpretation of the null findings is possible.

Yet even with Bayesian statistics, I am not convinced the findings will be conclusive. As the authors themselves mention, they tested participants who were equally proficient in LA+ and LA-. I believe that if the authors would like to disentangle the language vs memory vs methodological confounds while interpreting their null findings, they should have either included a control condition with linguistically imbalanced participants or at least should have included another math operation condition such as addition or subtraction.

Another statistical-related comment is that, surprisingly, the authors do not report any measurement of effect size. I think this information is crucial and must be provided.

In Ex.1 the authors blocked the task per language. However, with such a linguistically balanced sample, where participants are equally good in English and Spanish, I do not believe this was the right methodological choice if the authors would like to investigate the effect of LA. Generally, blocking leads to more efficient task performance. Therefore, if the authors would like to investigate the presence/absence of difference in semantic memory retrieval across LA+ and LA-, a randomized presentation would have been a better choice. If similar performances for LA+ and LA- would have been demonstrated using random presentation, this would have provided a piece of more convincing evidence. I believe that this is how Ex1A and Ex1B should have been structured – (a blocked vs random presentation) if the authors would like to address a methodological concern. Manipulating just the ISI between the experiments as it is done here, does not provide any additional insights.

Finally, the fact that there was a sample overlap of 12 participants (which is half of the sample) across the experiments could be a confound. The experiments are not that different from one another and the effects of interest might be affected (strengthened or weakened) by the repetition. Also from a statistical point of view, this is problematic. The authors should at least check if the performance for the overlapping participants is correlated.

RESULTS

  • Overall the ERP analyses lack information about which electrodes are analyzed as well as the justification forthe selection of specifically those electrodes.
  • In the same vein, the ERP figures do not display the topographies; theyare also difficult to read and to compare the time scales.
  • In the N400 analyses, the authors group the electrodes, but which and how many electrodes go in which hemisphere and based on what is unclear. More generally, the analyses of P300 and N400 are quite different from one another and there is no sound rationale why this was the case. For instance, when analyzing the N400 the authors conduct ANOVA, where one of the factors has 26 levels. Why are suddenly electrodes part of the analysis? Was there a lateralization hypothesis? If yes, why lateralization was not considered a factor when analyzing P300? In numerical cognition literature, lateralization patterns have been widely demonstrated and they can even differ based on the numerical format.
  • In addition, for a study investigating language, it would beinteresting to have both left and right hemisphere electrodes (at least in Supplementary).
  • Judging from figure 4, it seems that the components indeed follow a very similar pattern except for the last row (MiOc, presumably Medial Occipital part). This is rather bizarre.
  • Figure 5: It is puzzling that the congruent trials lead to more errors than the incongruent ones (is it an inversecongruency effect?), even more so given the faster reaction times for congruent than incongruent trials.

DISCUSSION

  • The conclusion contains strong, non-nuanced claims (i.e. lines 983-984). There is multiple evidence, for example insome of the papers cited in the reference that contradict those strong claims.

Minor points:

  • A minor recommendation is the organization of the paper. This might be a personal preference, but I find it quite difficult to have all the methodology of all experiments presented together and then all results together. This way it is difficult to follow up which methodology led to which results. I would rather keep it more classically – Ex1A – methods and results, Ex1B – methods and results.
  • The introduction is too long and does not read smoothly. I would propose to shorten it wherever possible and present the most relevant literature in a more summarized way. For instance, where very similar studies with slightly different methodologies and/or results have been discussed, a table or a diagram may be helpful.
  • Furthermore, there are a lot of results and it would be useful if the authors provide a small summary of their findings after each analysis.
  • L 744 The header should also contain “Experiment 2”
  • L 757 (cfr 754): Keep a common capitalization throughout the paper: Congruent or congruent, also followingin the results.

Reviewer 3 Report

The study is very interesting and explores a very interesting topic of arithmetic multiplication and ERP of Spanish\English bilinguals.

The main issue that I suggest the authors must address is the differences in the length of spoken numerals in Spanish and English (tres-three etc,...) in addition to the differences in syntactic structure of two digit numbers (especially 10-20).  I think that regarding these differences may affect multiplications decisions and ERPs on time axis. 

It is appreciated that the authors have balanced the spoken numbers into 450msec but the length and the number of letters\syllables must be considered  especially because if the differences between Spanish and English .